# Sustainable Development in Local Culture Industries: A Case Study of Taiwan Aboriginal Communities

**Cheng-Hsiang Yang [1], Yikang Sun [2] , Po-Hsien Lin [1] and Rungtai Lin [1,\*]**

[1] Graduate School of Creative Industry Design, National Taiwan University of Arts,
New Taipei City 22058, Taiwan; yjs.amo@gmail.com (C.-H.Y.); t0131@mail.ntua.edu.tw (P.-H.L.)
[2] College of Art & Design, Nanjing Forestry University, Nanjing 210037, China; sunyikang120110@hotmail.com
\* Correspondence: rtlin@mail.ntua.edu.tw

**Abstract:** Taiwan's indigenous communities have an abundance of unique cultures. Their service industries with local and foreign cultures have opened up distinct opportunities for sustainable development. Despite the enormous potential of aboriginal communities, particular attention should be given to ecology and sustainability. The traditional emphasis on craftsmanship and design is shifting to a new focus on the service industries and experimental design, which is not limited to the design of tangible products. Design concepts are now being applied to service industries that span several fields and are also being used to come up with systematic solutions for real-life problems. However, in the service industry, design experience must be used when introducing design concepts. The problem is how to shift from "High-tech" to "High-touch", for the aborigines are used to designing products at the usability level. This research proposes a model of experience design for use in aboriginal culture revitalization. Three different cases show how to apply the framework from experience design to local revitalization. Results show that the model can integrate the principles of sustainability into service industries and that it needs to be verified in future studies.

**Keywords:** sustainability; service innovation; experience design; revitalization; aborigine



## 1. Introduction

Due to the COVID-19 pandemic, Taiwan's aboriginal communities will likely be slow in recovering in comparison with other sectors of the tourism industry. Thus, an important task in the post-pandemic era will be to allocate the resources required to effectively revive the service industry in indigenous communities [1]. The poorer parts of the world that are most vulnerable to climate change are also most dependent on the service industry as a foreign exchange earner [2]. Tourism is acknowledged as a tool through which communities can achieve the United Nations' Sustainable Development Goals (SDGs) [3]. Economic, socio-cultural, and environmental impacts influence residents' support for service industry development [4]. However, despite the huge potential of the rural service industry, special attention must be paid to ecology and sustainability [5]. Indeed, the increasing importance being given to responsible service industry development in recent years has brought significant environmental, social, and economic benefits to rural communities worldwide [6]. Taiwan is no exception, where there is growing interest in both ecotourism and the cultural service industry conducted in aboriginal communities. At the same time, many aborigines see the service industry as a more responsible and sustainable way to both showcase and preserve their culture and ecological resources [1].

Previous studies detected relative problems in cultural heritage such as degradation of local culture industries, weak cross-sectoral linkages, ecosystem degradation, and loss of place integrity [7]. Sustainability emerged as the dominant practical and operational issue, while integration at different levels was the greatest challenge to service industry research [2]. By integrating the theoretical research on design, this study proposes a framework of experience design for the service industry in aboriginal communities, presenting

case studies showing how it is to be properly applied. In this way, service innovation will replace the tools of those involved in various aspects of aboriginal culture and sustainability.

For most Indigenous peoples, "sustainability" is the result of conscious and intentional strategies designed to secure a balance between human beings and the natural world and to preserve that balance for the benefit of future generations. Aboriginal people are also a source of sustainability strategies that can contribute to our collective well-being. Through ongoing communication and an understanding of traditional and environmental knowledge, education for a sustainable future can be achieved. Some successful cases, such as those in Australia and the United States, have inspired this study [8,9].

In short, the common goal is: Achieving sustainable development and promoting development cooperation. That is the core value of this study.

## 2. Literature Review

For aboriginal peoples, sustainability results in benefits for future generations. The concept is applied to secure a balance between human beings and the environment. Aborigines are a source of sustainability strategies that can contribute to service industries. Through education and communication of service innovation, sustainability can be achieved. Some concepts of service industries are needed to be changed as follows.

### 2.1. From High-Tech to High-Touch

Davis [10] proposed the technology acceptance model, which takes usability and ease of use as the main factors of design. Much earlier, Louis Sullivan (1856–1924) addressed the slogan of "form follows function" which became an important principle for examining the feasibility of design. In Taiwan, the development of manufacturing progress includes three stages, from original equipment manufacturing (OEM) to original brand manufacturing (OBM) [11]. In the OBM phase, designers have begun to integrate the emotional aspects of user experience into product design. Consequently, emotional design has become a key factor in product innovation. Thus, the emphasis of design has shifted "from Function to Feeling", "from Use to User", and finally "from High-tech to High-touch" [12–14]. In discussing the respective demands of consumers and product designers, Lin et al. [15] argue that in the evolution of the cultural and creative industries based on the 4C model (Creative, Cheerful, Collective, and Cultural), the emphasis of consumers has gradually shifted from physiological needs to aesthetic experience, while the emphasis of product designers has gradually shifted from function to experience, and is now in the process of shifting to culture, a process in which the 4C model is taking on a twin stage consisting of humanities and technology, beginning with creativity, and proceeding to enjoyment, choice, and culture, as shown in Figure 1 [11,15].

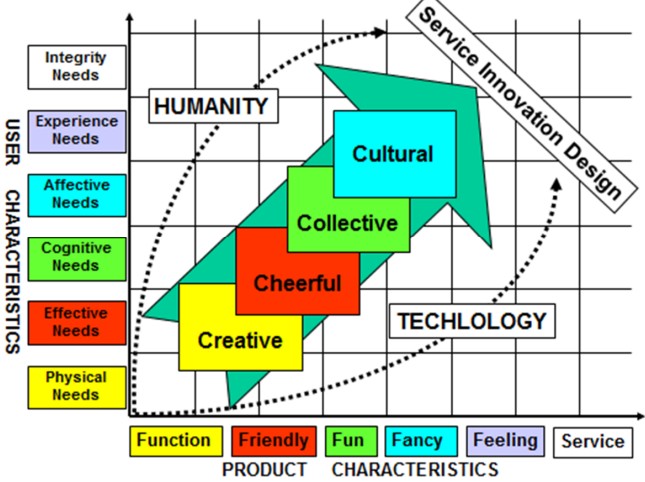

**Figure 1.** Framework of cultural creative industry's 4Cs. (Reprinted with permission from [11,15]. Copyright 2014, 2015 Lin, R. et al.).

Thus, due to the important changes in both consumers and manufacturers, increasing emphasis is being given to the emotional design stage of high-touch, aesthetics, and culture. Far from being limited to product design, this shift from high-tech to high-touch is also having a major impact on service innovation.

### 2.2. Local Revitalization in the Aboriginal Communities

Aborigines follow an understanding of the traditional belief system. In the aboriginal language, "Gaga", aboriginal ecological strategies advocate nature and human symbiosis that can contribute to our well-being in Taiwan [16]. The local environment can be used to increase interaction and understanding between locals and visitors [17], in the process of which local residents gain an enhanced sense of local identity. Moreover, local revitalization is multifaceted by nature and can include such elements as craft design, space arrangement, and service industry activities [18].

Sustainability has already become a vital development issue in aboriginal communities. It requires giving simultaneous consideration to the environment, society, and economy [19,20], since the benefits derived from each of these areas have a definite impact on a community's attitude toward development [4]. Service industries' development exerts mutual effects in economic, social-cultural, and environmental dimensions that change the host community's living experience [4].

Sustainable development can also serve as a practical stage for local revitalization, in that cultural heritage can become the driving force for sustainable development, and authenticity is a core value of culture [4,21]. Cultural exploration is the most important factor attracting tourists to aboriginal cultural festivals [22]. Sustainable development strategies are responsible for the historical-cultural context and the uniqueness of the community. Moreover, community-based approaches to development are the most effective and sustainable ones [23]. There are a variety of intangible elements of culture in aboriginal communities, including spoken stories, seasonal community activities, food, clothing, dance, way of life, and even memories [23].

For indigenous peoples, they follow their ancestors' ideas and have created a different lifestyle. As a result, many everyday items are very different from what we think. The culture of indigenous peoples and their artifacts are full of human nature, and how to add value through creativity, supplemented by elements of science and technology, both to preserve tradition and meet contemporary needs, is an urgent issue to be solved. In short: (1) From Function to Feeling, (2) From Use to User, and (3) From High tech to High touch (see Figure 2).

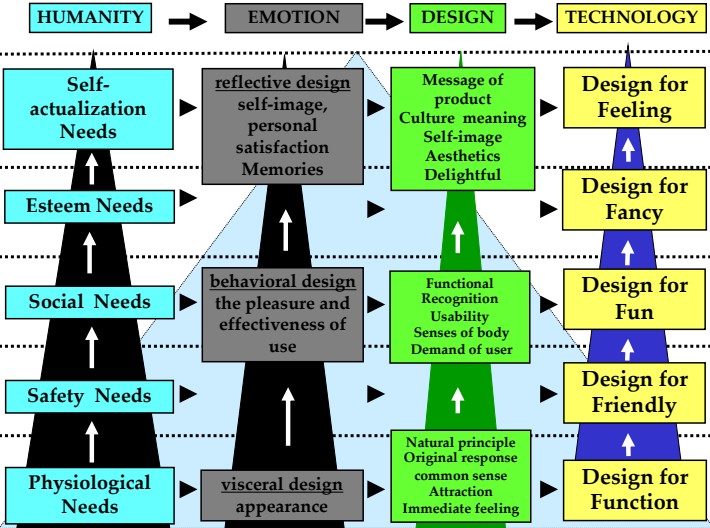

**Figure 2.** A framework of the relationship in humanity and technology. (Source: this study).

### 2.3. The Cultural and Creative Industries

Cultural product design is a process of rethinking or reviewing cultural features and then redefining them in order to design a new product to fit into society and to satisfy consumers with culture and aesthetics [24–26]. Hsu, Lin, and Lin [27] addressed a pattern for cultural product design which is (1) the idea is inspired from culture; (2) the idea must be formed into a product; (3) the product should be used in daily life; then (4) successes in branding are required, as shown in Figure 3.

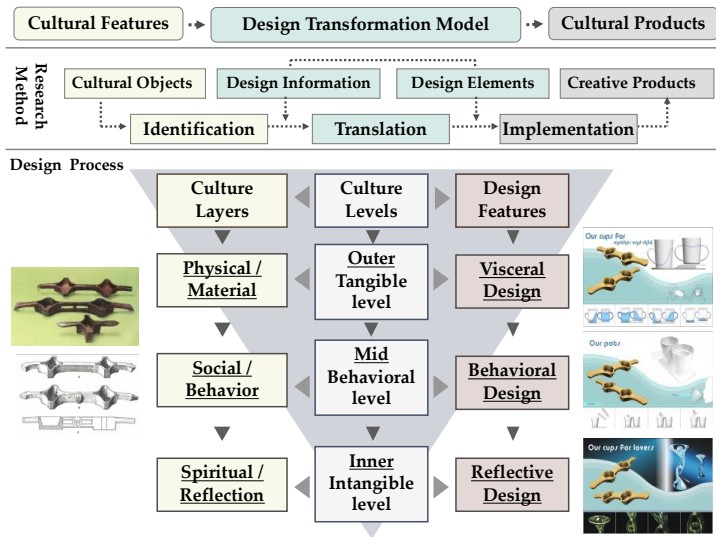

**Figure 3.** A model of the cultural and creative industries. (Adopted from [28]. Copyright 2016 Lin, R. et al.).

Hence, culture is the foundation of the cultural and creative industries, wherein cultural elements are creativity transformed into product designs that are integrated into daily life, finally establishing a brand image. In this study, culture is seen as a kind of way of life that groups people who have used a similar product and shaped a particular lifestyle.

Furthermore, Lin et al. [11,15] also proposed a model for cultural product design. Thus, the natural environment and local culture are the key elements of cultural products in the local service industry, wherein visitors engage in activities of local cultural rituals, in the process of which they gain a personal experience of the local culture [17]. Therefore, based on the previous studies [11,15,27], the cultural and creative industries are divided into a cultural phase consisting of nature and culture, and a production phase consisting of the creation of distinctive products relating to daily life, as shown in Figure 3.

### 2.4. Experience Design

Pine II and Gilmore formulated a model of the experience economy based on the value generated by economic activities, in which experience is regarded as a type of economic output, as well as an added-value commodity, such that the value generated by economic activities can be divided into four categories: commodities, goods, services, and experience [29]. The process by which a designer gradually transforms a text into a product can be said to consist of four steps: (1) setting a scenario; (2) telling a story; (3) writing a script, and (4) designing a product [25].

In the present study, "setting a scenario" is thought of as a process of turning creative inspiration into practical ideas; "writing a script" is thought of as the linking together of elements, which are arranged by the designer into a particular structure; and creative technology is used to shape these structured elements into a particular form. This study shows how the cultural and natural resources of the cultural phase can be extracted and used as components in writing a visual script on a particular theme. The resulting

experiential product is arranged in a perceptual context in such a way as to elicit a distinctive impression in the consumer, as shown in Figure 4.

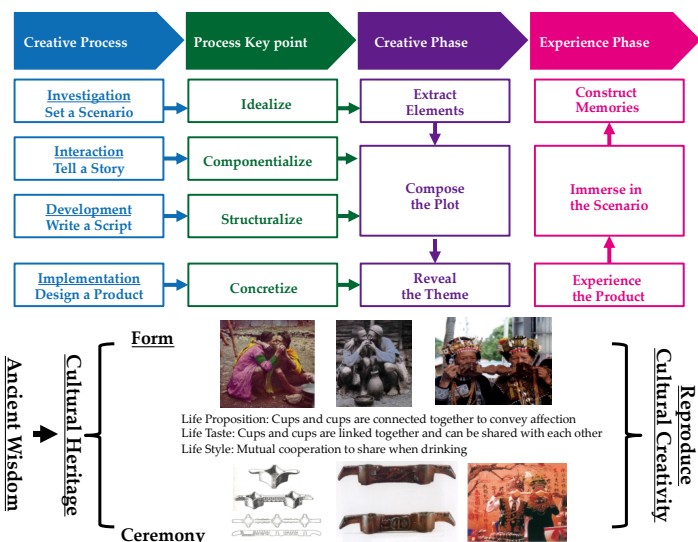

**Figure 4.** The stages of experience design. (Reprinted with permission from [28]. Copyright 2016 Lin, R. et al.).

## 2.5. Acculturation in Human–Culture Interaction

Acculturation plays an important role in cultural product design with embedded communication theory [30]. Communication is a process in which a message is encoded and conveyed via a particular medium to a recipient for decoding on various levels. In the case of a local festival, unanticipated and undesirable side effects include litter, noise pollution, traffic congestion, offensive behavior by tourists, and excessive commercialization, all of which tend to detract from the solemnity of what is supposed to be a sacred event. Even more problematic are external funding and the uneven distribution of the proceeds generated by the event, which can easily split a close-knit community into competing factions [31]. By contrast, in a sustainable approach to local development, equal or balanced emphasis is given to economic benefits, cultural preservation, and environmental protection [32].

In terms of culture, such an approach promotes the preservation and handing down of traditional skills, investing them with new values, both tangible and intangible, for residents and tourists alike [33]. Indeed, participation in cultural events is mainly motivated by the intrinsic value of the event itself, rather than by the pursuit of an instrumental value, such that authenticity is regarded as far more important than any entertainment value the event may offer [34]. Thus, one of the primary goals of local revitalization needs to be the preservation and handing down of local culture and traditional skills, as shown by the way in which the neglect of traditional culture or its inappropriate application often elicits a backlash from local residents. The past study appearance and physical settings of attractions were found to be the initial and most important indicators of authentic or inauthentic experiences. Other criteria for assessing the authenticity of heritage experiences include the presence of local culture and customs, constructed elements, commodification, and atmosphere [35]. In terms of economics, innovation is the key element for the feeling of freshness, all visiting the aboriginal sites are novelty-seekers in the aboriginal communities of Taiwan [36,37].

The media or agent by which the message of customer experience is transmitted can be staff, products, services, activities, or place, each of which has the ability to satisfy the consumer's requirements and to create a unique experience [38], which can be deepened and enhanced through the manipulation of sense perception and emotion, and by the use of the product itself. Past research has found that the main factors drawing visitors

to Taiwan's aboriginal communities are traditional architecture, handicrafts, traditional customs and rituals, and nostalgia for a bygone era [39].

As for the communication level, in his book *Emotional Design*, Norman [40] makes a detailed analysis of product design, in which he divides the creative design process into three levels: visceral, behavior, and reflection. For creativity, how to effectively communicate with local people, interest groups, etc., is a prerequisite for the subsequent series of work. Especially when it comes to indigenous peoples, it is necessary to further grasp the unique culture, customs, and lifestyles of indigenous peoples through communication. For indigenous peoples, their unique culture has created different ways of life. For designers, if they only look at the surface and draw conclusions, it is often easy to cause mistakes. Or rather, you cannot think in terms of past mindsets. Combining the above concepts and theories, this study further proposes a communication model suitable for cultural and creative design for indigenous peoples (Figure 5).

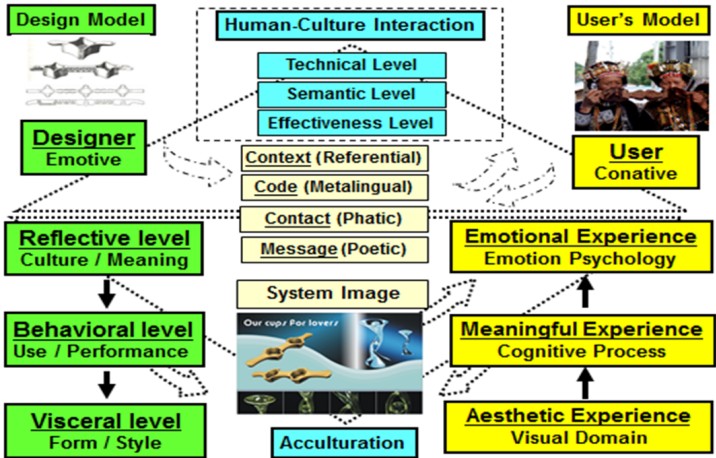

**Figure 5.** Acculturation for cultural and creative design. (Redraw from [30,41]. Copyright 2018 Gao et al.; 2009 Lin et al.).

## 3. Methodology

### 3.1. Conceptual Framework of Case Study

The case studies are suitable in the rich, real-world context in which the phenomena occur. The research question is broadly scoped, which gives the researcher more flexibility [42]. Moreover, the nascent theory proposes tentative answers to novel questions of how and why, often merely suggesting new connections among phenomena [43]. Hence, the study opted to explore the design context of indigenous culture sustainability development in the case study. The case study has more flexibility to cope with different villages' unique conditions.

Natural and cultural resources are the basis of the cultural and creative industries, and culture itself consists of the collective behavior formed by people interacting with and adapting to local natural conditions. The creative phase consists of extracting elements, writing a script, and presenting a theme. Next comes the communication phase, in which the designer can use either a formal or intentional approach in conveying his message. In a formal approach, the emphasis is on the use of local natural resources to convey a particular idea in the form of products, places, and festivals. In an intentional approach, the emphasis is on the use of communication and media to convey an abstract concept, such as creativity, authenticity, and sustainability. In the next phase, the designer brings his message to completion on the three levels of visceral, behavioral, and reflective. Once the message is received, the recipient processes it in terms of experience, emotion, memory, and impression. Next, the cultural and creative industry model is again used to present an actual product, one which relates to daily life, and thus has more significance for the consumer.

In the model, this study builds concentric circles on three levels which divide the cultural and creative industries, the experience, and the communication phase. In the outer

ring, this study uses the cultural and creative industries phase in the following way. Lin R. et al. propose four steps in inspiration from nature, forms culture, finds application in brand, and establishes daily life [28]. In the middle ring, this study uses the experience design phase in the following way. Lin et al. propose the following four steps: (1) setting a scenario, (2) telling a story, (3) writing a script, and (4) designing a product [25,28]. However, some elements' attributes are similar. So, this study simplifies some processes for the creation and experience phase. Finally, this study uses the communication phase following the inside ring. First, the communication phase divides into message, medium, and levels. The communication levels include visceral, behavior, and reflection from Norman. The communication medium includes ritual, workplace, and utensils [39]. The communication message includes authenticity, eco-friendly, and innovation [35–37].

In summary, this study combines several models obtained from the literature review, and then combines the motivation and goals of the research to construct the conceptual framework of experience design in local revitalization (Figure 6).

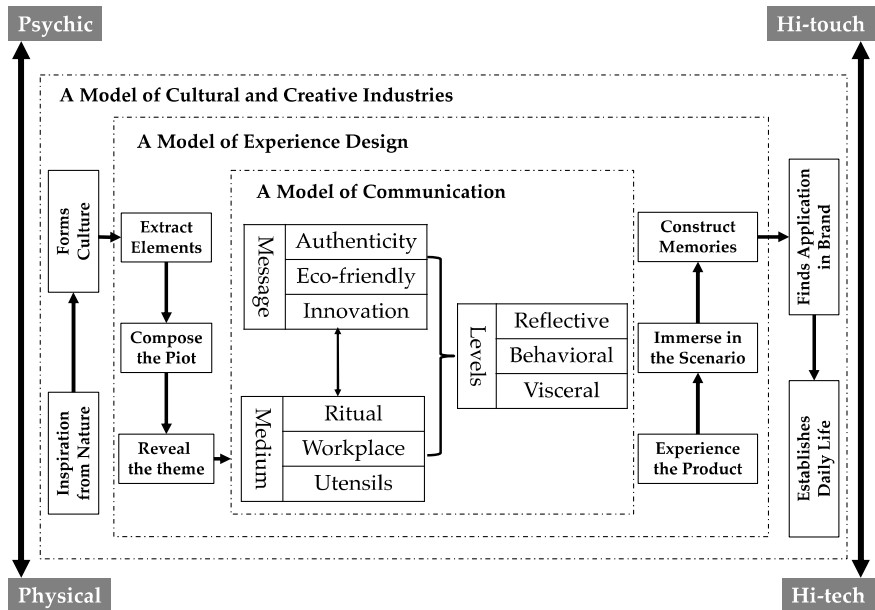

**Figure 6.** Conceptual framework of experience design in local revitalization. (Source: this study).

### 3.2. Overview of the Case

Despite ongoing outward migration and cultural erosion, Taiwan's aboriginal communities have managed to retain much of their distinctive cultural traditions, making them well suited for research on local revitalization. Taiwan presently has 16 officially recognized aboriginal tribes, all of which belong to the Austronesian language family. Among these, the largest is the Amis tribe, which is located on the eastern side of the island and whose members have traditionally made a living by fishing and agriculture. The Atayal tribe is spread out in the mountainous areas in the north and center of the island, and its members are noted for their skill in hunting and weaving.

The study opted for a case study approach to explore the design context of indigenous culture sustainability development. The case study has more flexibility to cope with different villages' unique conditions. This study is based on the medium element divided into three represented indigenous villages: (1) Ritual oriented is Fata'an village, (2) Workplace oriented is Raysina village, and (3) Utensils oriented is Ceroh village, all of which have sizable populations and have largely retained their cultural traditions.

So, the model presented in Figure 6 will be used to analyze the three villages and thus tap into the value of their cultural heritage and sustainable development.

## 4. Results: Case Analysis and Discussions

### 4.1. Traditional Utensils in the Village of Raysinay

Each of the three villages studied in this research emphasizes a different aspect of traditional culture. Raysinay (Shibi in Mandarin) gives special emphasis to the manufacture and use of traditional implements, especially textiles. This village is located in a deep river valley in a mountainous area.

In terms of the cultural-creative phase, the women of Raysinay are especially noted for producing textiles using traditional techniques and natural fibers and dyes. In terms of the creative phase, textiles serve as the extracted element; many Atayal textiles feature the "ancestor eye" motif, which serves as a kind of script; and the production of textiles using traditional dyes and techniques serves as the theme. In terms of the creative intension, visitors are shown how to make the dye by boiling local plants and given a chance to exercise their imaginations in making a tie-dye pattern, the final products of which are on display. In terms of the authenticity intension, visitors are shown how to collect natural ramie and how to weave it into rope, in the process of which they come to appreciate the wisdom and diligence which has been handed down through countless generations. In terms of sustainability intension, visitors participate in a sowing ritual and the collection of plants used for dyeing, thereby gaining insight into the traditional aboriginal emphasis on respect for nature and maintaining a balance between man and nature. In terms of the experiential phase, in the process of learning how to use local yams to prepare dye for tie-dyeing, visitors gain an understanding of the traditional Atayal belief system and mores (gaga), as embodied in the lore surrounding the ancestral eye. In terms of the production phase, inspired by the images created on the nearby cliffs by the movement of the sun, the villagers have created the Walkingsun textile brand, which features the primitive aesthetics of traditional Atayal motifs and dyes, as shown in Figure 7.

| Creative Phase | Produce Phase | | Communication Phase | | | Experience Phase | Industry Phase |
|---|---|---|---|---|---|---|---|
| | | | Innovation | Eco-Friendly | Authenticity | | |
| Forms Culture<br><br>Atayal | Extract Elements<br>Costume<br>Accessories | *Reflective*<br>Ritual<br>Workplace<br>Utensils | Display Fashion<br>Highlight Trait<br>Self-Fulfillment<br>Appreciated Together | Be Grateful to Land<br>Grateful for Everything<br>Pure Natural<br>Back to the Prime | Ancestors' Wisdom<br>Inherit Culture<br>Traveled Though Time<br>Conventional Method | Construct Memories<br>Commemorate Ancestors | Finds Application in Brand<br>Walkingsun |
| | Compose the Plot<br>Ancestor Eye | *Behavioral*<br>Ritual<br>Workplace<br>Utensils | Aesthetic Effectiveness<br>Unique Style<br>Immerse in Imagination<br>Stimulate Creativity | Respect Nature<br>Pray for the Harvest<br>Wild Plant<br>Esteem Origin | Follow Tradition<br>Follow the Custom<br>Experience in Person<br>Local Collection | Immerse in the Scenario<br>Gaga | |
| Begins with Nature<br><br>Fiber Dye | Reveal the Theme<br>Aboriginal<br>Weaving | *Visceral*<br>Ritual<br>Workplace<br>Utensils | Natural Dyeing<br>Boiled Dyeing<br>Dyeing Room<br>Tie-Dye Pattern | Primeval Forest<br>Seeding Ceremony<br>Hillside Grove<br>Vegetable Dyeing | Totem Knit<br>Twisted Cord<br>Wicker Workshop<br>Natural Ramie | Experience the Product<br>Yams Tie-Dye | Establishes Daily Life<br>Primitive Aesthetics |

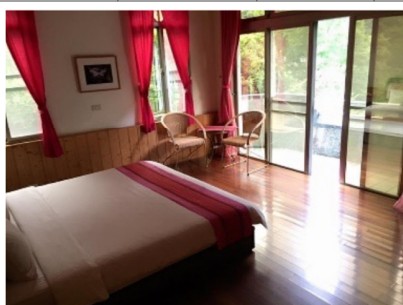
**Guest room featuring traditional textiles**

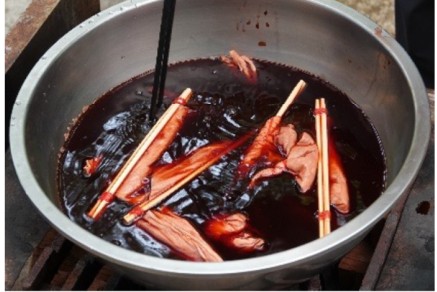
**Yam dye**

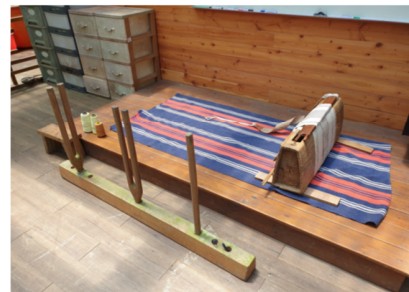
**Fabric featuring traditional Atayal motifs**

**Figure 7.** Traditional utensils of Raysinay. (Source: this study).

*4.2. Traditional Ritual in the Village of Fata'an*

Located next to the Matai'an Wetland Ecological Park in the East Rift Valley, the Amis village of Fata'an (Matai'an in Mandarin) is named after a type of legume that grows on trees, which was once a staple for the village.

In terms of the cultural-creative phase, in traditional Amis society, before participating in the customary coming-of-age ceremony, men are expected to be skilled in both hunting and fishing. The Amis of Fata'an have long used a technique for trapping fish known as palakaw, which has become a focus of the local service industry. In terms of the creative phase, "hunting and gathering" serves as the extracted element; "fishing in the river" serves as the script; and "fish trapping" serves as the theme. In terms of the creative intension, visitors are shown how to make eating utensils out of betel leaves, in the process of which they learn about the traditional aboriginal division of labor by which big jobs are tackled with ease. In terms of the authenticity intension, visitors collect stones from the river and are shown how to heat them for use in cooking fish, a technique which has been handed down for countless generations. In terms of the sustainability intension, visitors are shown how to construct a traditional fish trap out of natural materials, including bamboo and tree branches, in the process of which they come to appreciate the traditional Amis emphasis on living in harmony with nature. In terms of the experiential phase, visitors learn how to use the traditional Amis fish trap, in the process of which they gain a deeper appreciation for the traditional wisdom of Taiwan's aboriginals. In terms of the production phase, the traditional fish-trapping and stone-cooking techniques are adapted into a distinctive brand emphasizing simplicity and harmony with nature, as shown in Figure 8.

| Creative Phase | Produce Phase | Communication Phase | | | | | Experience Phase | Industry Phase |
|---|---|---|---|---|---|---|---|---|
| | | | | Innovation | Eco-Friendly | Authenticity | | |
| **Forms Culture** <br><br> Amis | **Extract Elements** <br> Fishing | *Reflective* <br> Ritual <br> Workplace <br> Utensils | | **Cooperative Learning** <br> Co-Creation <br> Life Innovation <br> Unexpectedness | **Self-Contentment** <br> Nature Conservation <br> Low Maintenance <br> Energy-Saving | **Trace the Source** <br> Simple and Unadorned <br> Inherit the History <br> Revere Ancestor | **Construct Memories** <br> Living <br> Wisdom | Finds Application in Brand <br> Fata'an |
| | **Compose the Plot** <br> River Fishing | *Behavioral* <br> Ritual <br> Workplace <br> Utensils | | **Teamwork** <br> Collaboration <br> Natural Workshop <br> Beyond Common Sense | **Self-Supporting** <br> Moderate Degree Fishing <br> Live Near a River <br> Draw on Local Resources | **Explore the Origin** <br> Original Flavour <br> Inheritance Territory <br> Historical Accumulation | **Immerse in the Scenario** <br> Primitive FIsherman | |
| **Begins with Nature** <br><br> Inland Wetland | **Reveal the Theme** <br> Diking for Fish | *Visceral* <br> Ritual <br> Workplace <br> Utensils | | **Natural Tableware** <br> Hand-Knit <br> Primary Forest <br> Betel Leaf Pot | **Set up Fishing Ground** <br> Build Habitat <br> Riverine Wetland <br> Bamboo Tube & Wooden Branch | **Search the Stone** <br> Stone Boiling <br> Stream Bank <br> Medical Stone | **Experience the Product** <br> Palakaw | Establishes Daily Life <br> Harmony with Nature |

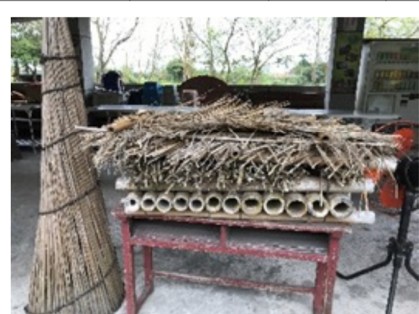

**Traditional three-level fish trap**

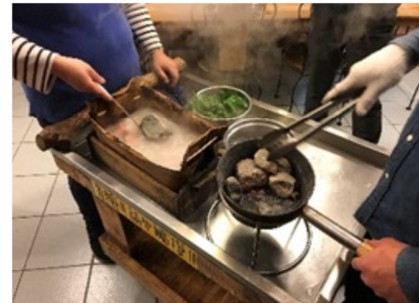

**Utensils made of betel leaves**

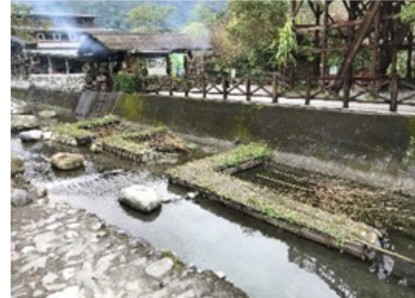

**Traditional fish trap in the river**

**Figure 8.** Rituals in Fata'an. (Source: this study).

*4.3. Local Color in Ceroh*

Also located in the East Rift Valley is the Amis village of Ceroh (Zhilou in Mandarin), which means "endless inflow of water." A huge footprint made by selectively removing rice stalks in a nearby paddy field was included in the popular film Beyond Beauty: Taiwan

from Above, placing Ceroh on the tourist map. In terms of the cultural-creative phase, the village is surrounded by open fields and clear-flowing rivers and is bisected by the Tropic of Cancer; the rich soil here is well suited for a variety of crops, and agriculture is an important part of the local economy and culture.

In terms of the creative phase, farming culture serves as the extracted element, the pristine natural environment serves as the script, and aboriginal agriculture serves as the theme. In terms of the creative intension, in addition to paddy art, the village is also known for its colorful festivals featuring traditional dance, all of which have made Ceroh a popular destination for domestic tourists. In terms of the authenticity intension, by participating in a traditional blessing ritual and by making a colorful "lovers' bag," visitors gain a taste of the ancient customs which have been handed down to the present day. In terms of the sustainability intension, visitors participate in the local harvest, thereby gaining an appreciation of the sustainable agricultural practices which have sustained the Amis for countless centuries. In terms of the experiential phase, visitors sample traditional Amis dishes served amongst the rice paddies, highlighting the village's natural farming techniques. In terms of the production phase, the village has launched the Mipaliu brand, highlighting a healthy lifestyle and the traditional Amis cooperative approach to farming, as shown in Figure 9.

| Creative Phase | Produce Phase | Communication Phase | | | Experience Phase | Industry Phase |
|---|---|---|---|---|---|---|
| | | | Innovation | Eco-Friendly | Authenticity | | |
| **Forms Culture**<br><br>Amis | **Extract Elements**<br>Plant Grain | *Reflective*<br>Ritual<br>Workplace<br>Utensils | **Enjoy Together**<br>Construct Relationship<br>Relieve Stress<br>Appreciate Together | **Ecology Forever**<br>Sparing Consuming<br>Symbiosis<br>Back to the Prime | **Return to the Hometown**<br>Deity Worship<br>Reminisce the Past<br>Promote Sence of Belonging | **Construct Memories**<br>Non-toxic | Finds Application in Brand<br>Mipaliu |
| | **Compose the Plot**<br>Pure Food | *Behavioral*<br>Ritual<br>Workplace<br>Utensils | **Open Mind**<br>Group Interaction<br>Relax Body<br>Stimulate Creativity | **Prevent Intervene**<br>Self Sufficient<br>Natural Framing<br>Cycle Utilization | **Distant Source**<br>Tranquillization of Mind<br>Long History<br>Self-Identity | **Immerse in the Scenario**<br>Rich Ground | |
| **Begins with Nature**<br><br>River Rice | **Reveal the Theme**<br>Aboriginal Farming | *Visceral*<br>Ritual<br>Workplace<br>Utensils | **Aboriginal Ceremony**<br>Dance with Other<br>Dining in Footprint Field<br>Colorful Rice | **Original Farming**<br>Farming Harvest<br>Natural Farmland<br>Leaf Tableware | **Etiquette of Life**<br>Chieftain Blessings<br>Sacrificial Temple<br>Lovers' Bag | **Experience the Product**<br>Rice Food | Establishes Daily Life<br>Healthcare |

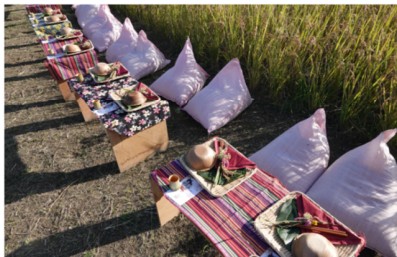

**Local dishes served inside a footprint in a paddy field**

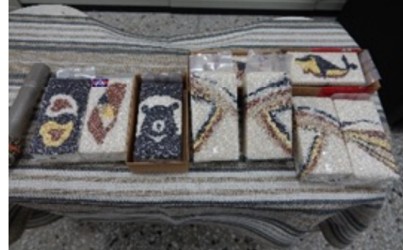

**Colorful rice**

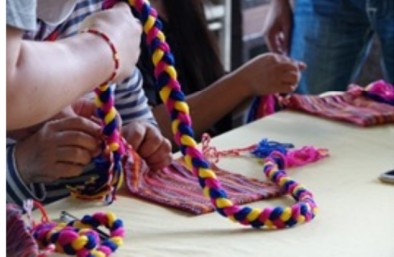

**Lover's bag**

**Figure 9.** The workshop stage in Ceroh. (Source: this study).

### 4.4. Discussions

Based on the model constructed in this study, economic, environmental, and cultural factors are examined in relation to sustainable development. In terms of economic factors, the focus is on using innovation to generate added value; in terms of culture, the focus is on using authenticity to create a sense of sacredness; and in terms of the environment, the focus is on using environmental awareness to generate a sense of connection with nature. The three aspects of economy, culture, and environment can be communicated through three different stages, viz., workshops based on economic and environmental considerations; rituals based on environmental and cultural considerations; and implements based on cultural and economic considerations, as shown in Figure 10.

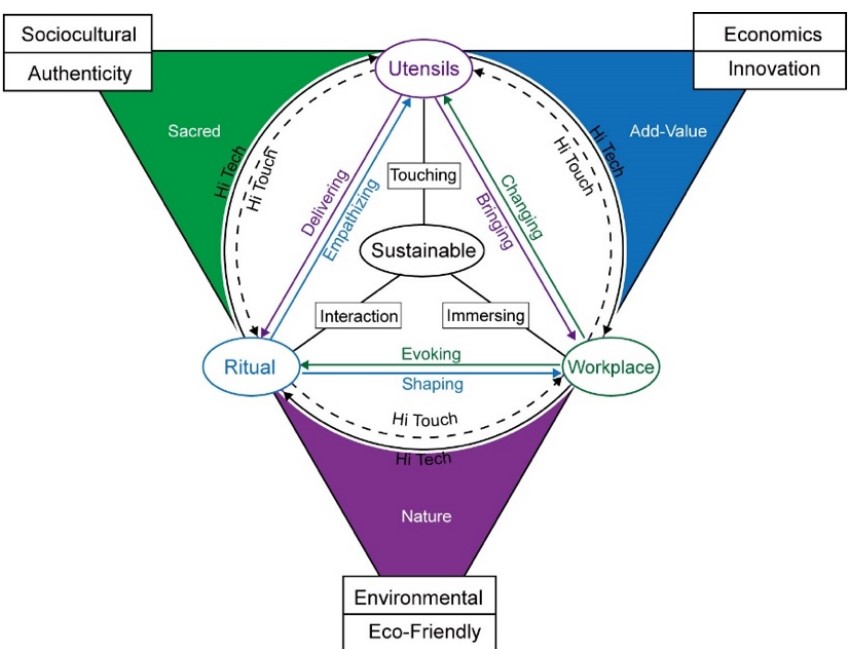

**Figure 10.** A model of sustainable experience design for local revitalization. (Source: this study).

It is believed that the local design elements originate from the indigenous people's lengthy culture and natural resources. Their designs are also in line with the aesthetic and experience needs of the current public to a certain extent. In using or viewing, people can better understand the culture and customs behind the objects. The indigenous villages developed different sustainable practices based on local natural conditions and unique culture. For example, (1) Atayal village has the best spinning and weaving in indigenous Taiwan because weaving is the key point in a woman's puberty rite. Hence, adult women need to be good at making a variety of life articles by themselves in Atayal villages; (2) Amis are the best fishers in indigenous Taiwan because they live around rivers and the beach. Therefore, they have developed a particular fishing ritual; (3) Ceroh village has flat topography and rich water, and later learned paddy farm skills from Hakka people. So, they excel in paddy farming more than other villages.

Regarding the role of the sacred in tourist experiences, this study argues that each ethnic group has its own unique culture and beliefs, many of which are mysterious elements. However, how the religious beliefs of the locals can be exploited as a tourist attraction is debatable. Sacred spaces and religion represent, from the tourist's point of view, a search for the authentic and an experience of the sacred. This is, then, tourism with spiritual connotations, which would alleviate the apparent ephemerality and lack meaning of everyday life [44]. The attitude of religious tourism in a sacred space is one of veneration and respect and seeks to have an experience that will put them in contact with the divinity and with a transcendental beyond [45]. The pilgrimage is now affected by new forms of motivation linked to the search for spirituality, authenticity, and cultural enrichment, resulting in new forms of tourism that provide an alternative to the traditional model [46]. It follows that when we talk about sacred spaces and tourism we must allow for different typologies of tourism; rather than confine ourselves to religious tourism, we must extend the field of study to other types, such as cultural or spiritual tourism [47–50]. For example, Richards [51] concludes that culture and tourism have a mutually beneficial relationship that can strengthen the attractiveness and competitiveness of local features.

In this study, indigenous culture is composed of many traditional mores and ceremonies. In addition, the long history and ancient myths have provided culture with strong sacredness. In indigenous culture, sacredness is primarily originally from nature's power and the souls of ancestors. In addition, the sacredness has a unique attraction and mystery for visitors' experience. When tourists take part in indigenous rituals, they recognize the

history and beliefs of indigenous language, dress, dance, and color. The attraction and mystery of sacredness for tourists is a burning question that requires an explicit and careful approach and deserves further study.

## 5. Conclusions and Suggestions

Based on the above case studies and discussions, the model constructed in this study can be used for further creative product development. Designing culture into products will become a design trend in culture industries. Product design has switched focus from usability and cognitive ergonomics to the affective aspects of user interactive experience. Therefore, this study is intended to study how to promote culture industries in the aboriginal community while nurturing sustainable development, the economic, environmental, social, cultural, and spiritual aspects of which were taken into account to balance three aspects of human–culture interaction. Case studies on three existing approaches to promoting products based on aboriginal culture are used to compare these three different design approaches, the results of which are used to construct a model of sustainable experience design for local revitalization.

In this study, three case studies are selected to verify the utility of the model as an approach to the study of culture industries and sustainable development. Due to differences in natural and cultural resources, each aboriginal village has adopted a different creative stage, which are named implement, ritual, and workplace, representing a shift from high-tech to high-touch as shown in the middle circle. In the implement stage, the emphasis is on physical contact with a particular object; in the ritual stage, the emphasis is on person-to-person interaction, mainly through meaningful normative behavior; and in the workshop stage, the emphasis is on the atmosphere conveyed by the place itself. Each stage has its own emphasis, and it is up to the designer to select the one which best fits the resources available in a particular local feature.

While each village needs to employ a variety of human–culture interactions to present what it has to offer potential visitors, it is also necessary to use a particular stage as the core value of sustainable development. Further studies are needed. For example, after the main stage is selected, the other two can be used in a subsidiary fashion to highlight the village's additional attractions. The designer might choose to emphasize high-tech functionality, bringing back memories with objects, and using implements to establish a connection with a place. For example, in constructing their traditional stone houses, the Paiwan tribe of southern Taiwan stack flat pieces of slate or shale in an arrangement which imitates the distinctive patterns seen on the hundred-pace snake—the tutelary deity of the Paiwan—and in laying the roof tiles of the same material, the larger pieces are used for the lower courses, and the smaller ones are installed higher up. Such traditional building techniques can be seen as a way of using the environment to evoke an emotional response. The aboriginal tribes such as Atayal, Saisiyat, and Paiwan have unique cultural features whose stages are better suited for different cultures that need to be studied in the future.

Relatively speaking, minorities or indigenous peoples have certain commonalities in the world, such as (1) the indigenous culture showing knowledge of human adaptation to different environmental conditions; (2) they belong to a relatively small minority in their countries or regions; and (3) their mystery and uniqueness attract more and more people. Therefore, we believe that the model of this study is an opportunity to be extended to other regions. However, since we are in Taiwan, limited to objective conditions, we can only take the indigenous people of Taiwan as an example to verify whether the relevant model is reliable.

**Author Contributions:** Conceptualization, R.L.; methodology, R.L.; writing—original draft preparation, C.-H.Y.; writing—review and editing, Y.S., P.-H.L. and R.L.; visualization, C.-H.Y. All authors have read and agreed to the published version of the manuscript.

**Funding:** This research received no external funding.

**Institutional Review Board Statement:** Not applicable.

**Informed Consent Statement:** Informed consent was obtained from all subjects involved in the study.

**Data Availability Statement:** Data sharing not applicable.

**Acknowledgments:** The authors would like to thank the Graduate School of Creative Industry faculty at the National Taiwan University of Arts for its support and valuable suggestions.

**Conflicts of Interest:** The authors declare no conflict of interest.

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
