# Peer review of "Sustainable Development in Local Culture Industries: A Case Study of Taiwan Aboriginal Communities"

_sustainability, doi:10.3390/su14063404_

Round 1

Reviewer 1 Report

The article is interesting for the object of its research and the presented conclusions. The study is closely linked to the perception of sustainability, harmony with nature and the environment. In addition to the analysis of the theoretical literature, in-kind research is performed by evaluating

Local businesses, the tools used for them and the festival. The transition from high technology to high sensitivity is to realize the vital experience passed down from generation to generation.

The scientific understanding of the article is explained by the illustrations provided, which systematize knowledge and experience and refine the main idea.Some notes:Can a similar study be carried out in other parts of the world, or is it unique to Taiwan? "Perhaps the review of the literature should be extended to a wider range of research.How to convey this experience using local design?

Author Response

Response to Reviewer 1  Comments

Article Title: sustainability-1613306- Sustainable Development in Local Culture Industries: A Case study of Taiwan Aboriginal Communities

The article is interesting for the object of its research and the presented conclusions. The study is closely linked to the perception of sustainability, harmony with nature and the environment. In addition to the analysis of the theoretical literature, in-kind research is performed by evaluating. Local businesses, the tools used for them and the festival. The transition from high technology to high sensitivity is to realize the vital experience passed down from generation to generation. The scientific understanding of the article is explained by the illustrations provided, which systematize knowledge and experience and refine the main idea. Some notes:

Thank you for your affirmation and encouragement of our research! Here are the answers to your comments:

Comment 1: Can a similar study be carried out in other parts of the world, or is it unique to Taiwan?

Response 1: Thank you for asking this question on a broader level. Relatively speaking, minorities or indigenous peoples have certain commonalities in the world, such as (1) the indigenous culture showing knowledge of human adaptation to different environmental conditions; (2) they belong to a relatively small minority in their countries or regions; and (3) their mystery and uniqueness attract more and more people. Therefore, we believe that the model of this study is an opportunity to be extended to other regions. However, since we are in Taiwan, limited to objective conditions, we can only take the indigenous people of Taiwan as an example to verify whether the relevant model is reliable.

Comment 2: Perhaps the review of the literature should be extended to a wider range of research. How to convey this experience using local design?

Response 2: Thank you for your comments! Based on the findings of the local investigation interview process, this study believes that the local design elements originate from the indigenous people’s lengthy culture and natural resources. Their designs are also in line with the aesthetic and experience needs of the current public to a certain extent. In using or viewing, people can better understand the culture and customs behind the objects.

Reviewer 2 Report

The topic is interesting and its approach is appropriate.

However, I think the article needs some changes and additions. So:

Subchapter 3.1.

-Present more clearly the connection between the theoretical sources that you mentioned before and the model that you propose;

-I suggest you enlarge figure 5, its content is difficult to read;

-Present more clearly the connection between the proposed model and the case studies

-Explain why you opted methodologically for the case study, why it is appropriate for the research topic

Chapter 4

-This chapter is about research results, I suggest you rename it just like that: Results

-Check the titles of subchapters 4.1 and 4.2: both are about the village of Raysinay?

-The presentation of the case studies is too descriptive, please comment on the significance of the results obtained, in Results:

-I suggest you move lines 330-350 from Conclusions to Results and add to the Conclusions only the summary of these lines, to link to the other ideas there.

-Show what you consider to be the connection between local peculiarities and the options of locals for the development of different sustainable practices.

-It is not clear what the role of the sacred is in tourist experiences, please explain.

Author Response

Response to Reviewer 2  Comments

Article Title: sustainability-1613306- Sustainable Development in Local Culture Industries: A Case study of Taiwan Aboriginal Communities

The topic is interesting and its approach is appropriate. However, I think the article needs some changes and additions. So:

Thank you for your recognition and encouragement of our article! Regarding your comments, here are our responses and changes:

Comment 1: Subchapter 3.1.

-Present more clearly the connection between the theoretical sources that you mentioned before and the model that you propose;

-I suggest you enlarge figure 5, its content is difficult to read;

-Present more clearly the connection between the proposed model and the case studies;

-Explain why you opted methodologically for the case study, why it is appropriate for the research topic.

Response 1: About Subchapter 3.1.

  • -This study proposes model divided into four parts, including “2.1. Hi-tech and Hi-touch”, “2.3. Culture and Creative Industries”, “2.4. Experience Design”, and “2.5 A model of Communication”. Finally, this study integrates four parts to build a Conceptual Framework of experience design in local revitalization.
  • -Figure 5 has been enlarged for readability. Based on the reviewer’s reminder, we further checked the other figures to make sure that all the images were of the right size and that the content in the figures was clear and legible.

It should be noted that the number of the figure has not been validly verified in the first version. We have made changes. Therefore, figure 5 referred to by the reviewer was corrected to figure 6 in the revised version.

  • -This study is based on the medium element divided into three oriented and opted three represented indigenous villages. Includes: (1) Ritual oriented is Fata’an village, (2) Workplace oriented is Raysinay village, and (3) Utensils oriented is Ceroh village.
  • -The study opted case study to explore the design context of indigenous culture sustainability development. The case study has more flexibility to cope with different villages' unique conditions.

Comment 2: Chapter 4

-This chapter is about research results; I suggest you rename it just like that: Results.

-Check the titles of subchapters 4.1 and 4.2: both are about the village of Raysinay?

-The presentation of the case studies is too descriptive, please comment on the significance of the results obtained, in Results:

-I suggest you move lines 330-350 from Conclusions to Results and add to the Conclusions only the summary of these lines, to link to the other ideas there.

-Show what you consider to be the connection between local peculiarities and the options of locals for the development of different sustainable practices.

-It is not clear what the role of the sacred is in tourist experiences, please explain.

Response 2: Thank you for reviewing these details! We will review the full text again to ensure consistency to avoid misunderstandings between reviewers and readers.

  • -Thank you for your advice. We changed the title of Chapter 4 to “Results: Case Analysis and Discussions”.
  • -Thank you for such a meticulous review! We have modified the titles of subchapters 4.1 and 4.2.
  • -Thanks for the reminder! Appropriate content has been adjusted and completed.
  • -The indigenous village developed different sustainable practices based on local natural conditions and unique culture. Such as: (1) Atayal village is the best spin and weave in Taiwan indigenous because the weaving skill is the key point in a woman’s puberty rite. The adult woman makes a variety of life articles by himself in Atayal village; (2) Amis is the best fisher in Taiwan indigenous because they live around river and beachside. Therefore, they develop a particular fishing ritual.
  • -The role of the sacred is in tourist, this is our point of view: Each ethnic group has its own unique culture and beliefs, many of which are more mysterious elements. For the purposes of this study, the indigenous culture is composed of many traditional mores and ceremonies. In addition, the long history and ancient myths provided to culture was strong sacredness. In indigenous culture, the sacredness is primarily original from nature’s power and ancestry soul. And the sacredness has a unique attraction and mystery for visitors’ experience.

Round 2

Reviewer 2 Report

Dear authors,

The clarifications / additions requested in the previous review are for the improvement of the article, not for the reviewer.

In this context, please insert them in the article.

In some cases, further clarification is useful:

-Present more clearly the connection between the theoretical sources that you mentioned before and the model that you propose (In the article: In the model we propose we use source X in the following way ...., source Y in the following way ....)

-Present more clearly the connection between the proposed model and the case studies (In the article: We use the model to to highlight / present / analyze…)

-Explain why you opted methodologically for the case study, why it is appropriate for the research topic (In the article,  more consistent than The case study has more flexibility to cope with different villages' unique conditions, possibly supported by bibliographic references)

-Show what you consider to be the connection between local peculiarities and the options of locals for the development of different sustainable practices (In the article: insert good ideas from the response to the review in the article).

-It is not clear what the role of the sacred is in tourist experiences, please explain (In the article: insert your ideas from the response to the review in the article).

As for the figures, I think they are a little too big this time.

Author Response

Response to Reviewer 2  Comments (Round 2)

Article Title: sustainability-1613306- Sustainable Development in Local Culture Industries: A Case study of Taiwan Aboriginal Communities

Comments

Dear authors,

The clarifications / additions requested in the previous review are for the improvement of the article, not for the reviewer.

In this context, please insert them in the article.

In some cases, further clarification is useful:

-Present more clearly the connection between the theoretical sources that you mentioned before and the model that you propose (In the article: In the model we propose we use source X in the following way ...., source Y in the following way ....)

-Present more clearly the connection between the proposed model and the case studies (In the article: We use the model to highlight / present / analyze…)

-Explain why you opted methodologically for the case study, why it is appropriate for the research topic (In the article, more consistent than The case study has more flexibility to cope with different villages’ unique conditions, possibly supported by bibliographic references)

-Show what you consider to be the connection between local peculiarities and the options of locals for the development of different sustainable practices (In the article: insert good ideas from the response to the review in the article).

-It is not clear what the role of the sacred is in tourist experiences, please explain (In the article: insert your ideas from the response to the review in the article).

As for the figures, I think they are a little too big this time.

Response

Thank you again for your rigorous and meticulous review of our article, not only for further pointing out the problems that still exist but also for giving detailed tips.

Based on your comments, we have added the content involved to the appropriate positions in the article and marked them in red. For your convenience, they are located as follows:

-Present more clearly the connection between the theoretical sources that you mentioned before and the model that you propose.

Line 237~252.

-Present more clearly the connection between the proposed model and the case studies.

Line 264~272.

-Explain why you opted methodologically for the case study, why it is appropriate for the research topic (In the article, more consistent than the case study has more flexibility to cope with different villages’ unique conditions, possibly supported by bibliographic references).

Line 216~222 and Line 523~526.

-Show what you consider to be the connection between local peculiarities and the options of locals for the development of different sustainable practices.

Line 361~372.

-It is not clear what the role of the sacred is in tourist experiences, please explain.

Line 373~379.

-As for the figures, I think they are a little too big this time.

We re-examined the figures in the full text. Adjustments were made for some of the poor resolutions. In addition, we have made technical treatment of the size of the figure according to the format requirements of the journal to minimize unnecessary blank space. If the article is finally accepted, we will also assist the editor in adjusting the figure again according to the requirements of the publication.

About the modification of other parts. In addition to a full review of the issues you mentioned, we have once again reviewed the full text. Some unclear or erroneous formulations have been modified. These sections are also marked in red.

Round 3

Reviewer 2 Report

Dear authors,

Please review the line 373 (”Regarding the role of the sacred is in tourist experiences...”)
I still think it would be useful to explain in more detail how the religious beliefs of the locals can be exploited as a tourist attraction. This is an idea that requires a careful explicit approach.

Author Response

Response to Reviewer 2  Comments (Round 3)

Article Title: sustainability-1613306- Sustainable Development in Local Culture Industries: A Case study of Taiwan Aboriginal Communities

Comments

Dear authors,

Please review the line 373 (”Regarding the role of the sacred is in tourist experiences...”)

I still think it would be useful to explain in more detail how the religious beliefs of the locals can be exploited as a tourist attraction. This is an idea that requires a careful explicit approach.

Response

Dear reviewer,

Thank you very much for your comments! Your opinion of the article not only makes the article more perfect but also brings us a lot of inspiration!

Based on your latest comment, we further review the literature related to it and then present it based on the purpose and findings of this study. The modified or added content is on “Line 375~389” and “Line 394~397” respectively.

There are a total of 8 references for additional content that are listed in the reference entries. (Line 545~560)
